# Adherence to Oral Chemotherapy in Acute Lymphoblastic Leukemia during Maintenance Therapy in Children, Adolescents, and Young Adults: A Systematic Review

Xiaopei L. Zeng [1,2,*], Mallorie B. Heneghan [3] and Sherif M. Badawy [1,2]

1    Department of Pediatrics, Northwestern University Feinberg School of Medicine, Chicago, IL 60611, USA; sbadawy@luriechildrens.org
2    Division of Hematology, Oncology and Stem Cell Transplantation, Ann & Robert H. Lurie Children's Hospital of Chicago, Chicago, IL 60611, USA
3    Division of Hematology, Oncology and Stem Cell Transplantation, Department of Pediatrics, University of Utah, Salt Lake City, UT 84132, USA; mallorie.heneghan@hsc.utah.edu
*    Correspondence: xzeng@luriechildrens.org; Tel.: +1-312-227-5090; Fax: +1-312-227-9756

**Abstract:** Acute lymphoblastic leukemia (ALL) is the most common malignancy in children and young adults. Treatment is long and involves 2–3 years of a prolonged maintenance phase composed of oral chemotherapies. Adherence to these medications is critical to achieving good outcomes. However, adherence is difficult to determine, as there is currently no consensus on measures of adherence or criteria to determine nonadherence. Furthermore, there have been few studies in pediatric B-ALL describing factors associated with nonadherence. Thus, we performed a systematic review of literature on oral chemotherapy adherence during maintenance therapy in ALL following PRISMA guidelines. Published studies demonstrated various objective and subjective methods of assessing adherence without generalizable definitions of nonadherence. However, the results of these studies suggested that nonadherence to oral maintenance chemotherapy was associated with increased risk of relapse. Future studies of B-ALL therapy should utilize a uniform assessment of adherence and definitions of nonadherence to better determine the impact of nonadherence on B-ALL outcomes and identify predictors of nonadherence that could yield targets for adherence improving interventions.

**Keywords:** leukemia; lymphoblastic leukemia; acute lymphoblastic leukemia; ALL; adherence; compliance; pediatric; children; adolescents; young adults; AYA; oral chemotherapy; cancer

## 1. Introduction

Acute lymphoblastic leukemia (ALL) is the most common cancer affecting children and adolescents [1]. The 5-year survival rate is approaching 90% after treatment with multi-agent chemotherapy [2,3]. Treatment typically lasts 2–3 years, and begins with up to 9 months intensive chemotherapy followed by a prolonged low-intensity maintenance phase that lasts for the remainder of therapy [4]. Maintenance phase is predominated by an oral chemotherapy regimen consisting of daily 6-mercaptopurine (6-MP) or, less commonly, daily 6-thioguanine (6TG), weekly methotrexate (MTX), and intermittent oral steroid bursts, all given at home by patients or caregivers [5]. The long duration of maintenance therapy has been shown to reduce odds of relapse [5]; however, it requires adequate adherence to the prescribed regimen.

Oral chemotherapy has many perceived benefits compared to intravenous (IV) chemotherapy, including greater flexibility, fewer interruptions to usual routines, fewer trips to the hospital, and reduced stress [6,7]. However, administration of medications at home provides new challenges, as the burden of medication administration is shifted from provider to patient. One of the most notable challenges is medication adherence [8].

Medication adherence can be defined as taking medications exactly as prescribed by a medical provider. It includes taking the right medication at the right dose at the right time, consistently [9]. Across oncologic conditions, adherence to oral chemotherapy is a challenge, with widely variable adherence rates of 17–100% [8].

Given the importance of maintenance phase therapy in preventing relapse of ALL, our primary objective was to systematically review the literature on oral chemotherapy adherence during maintenance therapy in pediatric ALL with a particular focus on individual medication adherence rates and the overall prevalence of nonadherence. The secondary aims were to explore the possible correlates of nonadherence and the relationship between oral chemotherapy adherence and health outcomes, including event-free survival and relapse.

## 2. Methods

We followed the guidelines for Preferred Reporting Items for Systematic Reviews and Meta-Analyses (PRISMA) [10] (Figure 1).

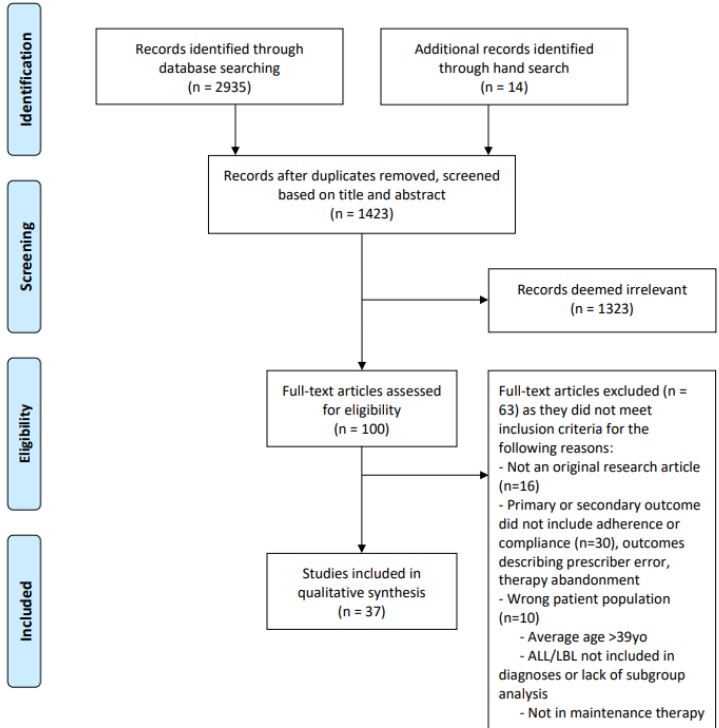

**Figure 1.** PRISMA flow diagram delineating article selection methodology for inclusion in systematic review.

### 2.1. Article Retrieval

The search strategy involved looking for all articles relating to adherence or compliance in pediatric and young adult acute lymphoblastic leukemia. A medical librarian conducted a literature search in the following databases: PubMed, EMBASE, Cumulative Index to Nursing and Allied Health Literature (CINAHL), PsycINFO, and Cochrane. No year limits or language limits were applied. Specific keywords were searched, including leukemia, patient compliance, medication adherence, treatment refusal, treatment barriers, treatment dropouts, treatment compliance, and search terms relating to child, pediatric, adolescent, and young adult. An age limit applied, from 0–39 years old. Additional articles were added during the review process, through hand search of PubMed, using the following keywords: adherence, compliance, leukemia, and maintenance chemotherapy. Two independent reviewers (MH and XZ) assessed titles, abstracts, and full-text articles against eligibility criteria. Disagreements were resolved by discussion or consultation with a senior author (SB). An updated literature search on PubMed, using keywords from previous hand search,

was completed in December 2022. Two independent reviewers (XZ and SB) assessed titles, abstracts, and full-text articles against eligibility criteria.

### 2.2. Article Selection

Inclusion criteria for this review were as follows: (1) children, adolescent, and young adults (0–39 years old) with acute lymphoblastic leukemia/lymphoma in remission, on oral maintenance chemotherapy, (2) original full-text research articles, (3) medication adherence or nonadherence as the primary or secondary outcome. Adherence, in this review, was defined as the consistency with which individual patients took prescribed medication. Exclusion criteria included (1) mean or median age of participants greater than 39 years old, (2) focus on non-maintenance phase of therapy, (3) inclusion of multiple malignancies without subgroup analysis of ALL/LBL, (4) focus on prescriber error or abandonment of therapy, and (5) no full-text article available.

### 2.3. Data Extraction and Synthesis

A standardized form was used for data extraction. Data items in the form included the following: first author's name, publication year, country, study design and aims, participants' age and sex, race or ethnicity (if reported), sample size, duration of study and long-term follow-up, maintenance medication evaluated, adherence measures, adherence rates, definition and prevalence of nonadherence, interventions (if applicable), correlates of nonadherence, and health outcomes, including event-free survival and relapse. Each article included in the review was evaluated for quality of evidence using the Cochrane GRADE approach (Grades of Recommendation, Assessment, Development, and Evaluation) [11]. Disagreements were resolved by discussion or consultation with a senior author. Data were analyzed and summarized qualitatively. Our primary outcome measure was medication adherence rate and prevalence of nonadherence. Secondary outcome measures included clinical outcomes (survival, relapse).

## 3. Results

### 3.1. Literature Search

A total of 2935 citations were retrieved; 14 additional references were identified through hand search. After duplicates were removed, 1423 citations remained; 100 full-text articles were assessed for eligibility. A total of 37 articles met all inclusion criteria. This process was outlined in the PRISMA flow chart, showing reasons for exclusion of full-text articles (Figure 1).

### 3.2. Study Characteristics

Table 1 summarizes characteristics of the 37 publications included in this review [12–48]. The 37 publications represented 28 unique studies published between 1979 and 2021. The majority of the publications included exclusively patients diagnosed with ALL (*n* = 30). A few publications including multiple malignancies (*n* = 3) had ALL/LBL as the largest subgroups, ranging from 49–94%. Most had a median or mean age of less than 10 years old (*n* = 18), with only 7 publications focusing specifically on adolescents and young adults. Adherence assessments of thiopurine (predominantly 6MP) were most common (*n* = 28), followed by methotrexate (*n* = 5), combination (*n* = 3), and pulse steroids (*n* = 2). Most publications were from the United States and Canada (*n* = 18) or Western Europe (*n* = 12). Several (*n* = 7) were from middle income countries in South America and Africa. There were no publications from Asian countries. Prospective cohort studies or analysis of adherence data from randomized control trials (*n* = 23) were the most common study designs, with only 4 that analyzed adherence-promoting interventions. Sample sizes ranged from 16 to 1194; approximately 50% had fewer than 100 participants (*n* = 19), while only 4 publications reported on studies with greater than 700 participants. Duration of data collection ranged from a single time point to 2 years (the entire duration of maintenance therapy). Additionally, 7 of 8 articles reporting long-term follow-up reported a follow-up of greater than 3 years.

**Table 1.** Study characteristics.

| Source, Year (Country) | STUDY DESIGN | Study Aim | Medication | Sample Size N (*Subgroup n*) | Age at Study Entry, Mean or Median (SD or Range) in Years | Study Length *Follow Up Duration* | Grade |
|---|---|---|---|---|---|---|---|
| Alsous et al., 2017 [12] (Jordan) | Cross-sectional; single center ≤19 year old, ALL | Assess adherence to 6MP; identify factors influencing adherence | 6MP | 52 | 8.9 (4.4) | Single time point | VL |
| Bhatia et al., 2012 [13] (USA/Canada) | Prospective observational; multicenter ≤21 year old, ALL [a] | Assess adherence to 6MP, impact on relapse, relation to ethnicity (Hispanic vs. non-Hispanic white) | 6MP | 327 | 4 (1–19) | Data collected over 6 months *Median (range): 3.7 (0.4–8.8) years* | M |
| Bhatia et al., 2014 [14] (USA/Canada) | Prospective observational; multicenter ≤21 year old, ALL [a] | Assess adherence to 6MP, impact on relapse, relation to ethnicity (African Americans/ Black, Asian, non-Hispanic white) | 6MP | 298 | 6 (2–20) | Data collected over 5 months *Median (range): 5 (0.07–9.1) years* | M |
| Bhatia et al., 2015 [15] (USA/Canada) | Prospective observational; multicenter ≤21 year old, ALL [a] | Evaluate intra-individual variability of 6MP on relapse risk | 6MP | 742 (*Adherence data n = 470*) | 6 (2–21) | Data collected over 6 months. *Median (range): 5.2 (0.07–9.4) years* | M |
| Bhatia et al., 2020 [16] (USA/Canada) | Randomized control trial with intervention; multicenter ≤21 year old, ALL | Determine if multicomponent intervention (text + direct supervision + education) will increase adherence to 6MP compared to education alone | 6MP | 444 (*Intervention n = 230; Education alone n = 214*) | 8.1 (IQR 5.3–14.3) | Data collected over 28 days for baseline, intervention period of 16 weeks | H |
| Davies et al., 1993 [17] (UK) | Prospective observational; single center "Children", ALL | Assess adherence to 6MP | 6MP | 35 | NR (NR) | Data collected at least 2 time points, unknown duration | VL |
| De Oliveira et al., 2004 [18] (Brazil) | Prospective observational; single center <18 year old, ALL [b] | Assess adherence to 6MP | 6MP | 39 | 4.8 (1.5–16.3) | Data collected over entirety of maintenance *Median (range): 5.25 (1.38–6.9) years* | L |

**Table 1.** *Cont.*

| Source, Year (Country) | STUDY DESIGN | Study Aim | Medication | Sample Size N (Subgroup n) | Age at Study Entry, Mean or Median (SD or Range) in Years | Study Length *Follow Up Duration* | Grade |
|---|---|---|---|---|---|---|---|
| De Oliveira et al., 2005 [19] (Brazil) | Prospective observational; single center <18 year old, ALL [b] | Assess adherence to 6MP | 6MP | 73 | 4.0 (1.2–16.3) | Data collected over entirety of maintenance *Median (range): 4.75 (1.33–8.5) years* | L |
| Farberman et al., 2021 [20] (Argentina) | Cross-sectional; multicenter 0–17 year old, ALL/LBL | Assesses adherence to oral chemotherapy and beliefs | Oral maintenance | 203 (ALL *n* = 163) | NR (NR) *0–2 (2.5%); 2–6 (34.5%); 6–11 (36.6%); 11–18 (27.1%)* | Single time point | L |
| Hawwa et al., 2009 [21] (Northern Ireland) | Prospective observational; single center "Children", ALL | Develop method to prospectively assess adherence to 6MP | 6MP | 19 | 10 (3–17) | Data collected over 6 months | L |
| Heneghan et al., 2020 [22] (USA) | Cross-sectional; single center 1–27 year old (parent of 1–18 year old, patient 12–24 year old), ALL | Assess parent and patient reported adherence and barriers to adherence | 6MP | 57 families (*Patient n* = 16; *Parent n* = 49) | Patient participants: 17 (IQR 16–19) Patient reported on by parents: 6 (IQR 5–10) | Single time point | L |
| Hoppmann et al., 2021 [23] (USA/Canada) | Prospective observational; multicenter ≤21 year old, ALL [a] | Develop risk prediction model for 6MP nonadherence | 6MP | 407 | 7.7 (4.4) | Data collected over 6 months | M |
| Isaac et al., 2020 [24] (USA) | Prospective observational; multicenter 7–19 year old, ALL/LBL [c] | Assess ethnic differences in parent and child social problem-solving abilities and impact on 6MP adherence | 6MP | 139 | 12.3 (3.4) | Data collected over 15 months | M |
| Jaime-Perez et al., 2009 [25] (Mexico) | Prospective observational; single center ≤15 year old, ALL | Assess adherence to MTX | MTX | 49 | 8 (5–15) | Data collected over 6–7 months | L |

**Table 1.** *Cont.*

| Source, Year (Country) | STUDY DESIGN | Study Aim | Medication | Sample Size N (Subgroup n) | Age at Study Entry, Mean or Median (SD or Range) in Years | Study Length *Follow Up Duration* | Grade |
|---|---|---|---|---|---|---|---|
| Kato et al., 2008 [26] (USA/Canada/ Australia) | Randomized control trial; multicenter 13–29 year old, multiple malignancies (subgroup ALL/LBL on 6MP) | Determine if video-game intervention will increase adherence and alter other behavioral outcomes in adolescents with malignancies | 6MP | 375 (*ALL n = 152; 6MP n = 54*) | NR (13–29) *Adolescent* [13–16] *(66%)*; *Young adult* [17–29] *(34%)* | Data collected over 3 months | M |
| Khalek et al., 2015 [27] (Egypt) | Prospective observational; single center "Children", ALL | Assess adherence to 6MP | 6MP | 129 | 6 (1.6–16.1) | Data collected over 15 months | L |
| Kremeike et al., 2015 [28] (Germany) | Prospective observational; single center ≤18 year old, ALL | Assess factors influencing adherence during maintenance therapy | 6MP, MTX | 33 | 8.2 (1–16) | 1–3 time points, unknown time-period | VL |
| Kristjansdottir et al., 2021 [29] (Denmark) | Retrospective; multicenter 18–45 year old, ALL | Assess adherence to 6MP and association with survival in young adults | 6MP | 62 (*Adherence data n = 51*) | 24.2 (IQR 19.4–33.5) | Data extracted over 11 year period *Median (range): 4.1 (0.6–10.7)* | M |
| Lancaster et al., 1997 [30] (UK) | Cross-sectional cohort; multicenter "Children", ALL | Assess interpatient variability at standardized dose of 6MP | 6MP | 496 | NR | Single time point | VL |
| Landier et al., 2017 [31] (USA/Canada) | Prospective observational; multicenter ≤21 year old, ALL [a] | Comparison of self-reported adherence to electronic monitoring; identify predictors of overreporting | 6MP | 416 | 7 (2–20) | Data collected over 4 months | M |
| Landier et al., 2017 [32] (USA/Canada) | Prospective observational; multicenter ≤21 year old, ALL [a] | Assess 6MP ingestion habits and impact on adherence and relapse | 6MP | 441 | 6 (2–20) | Data collected over 6 months *Median (range): 6.1 (0.8–11) years* | M |

| Source, Year (Country) | STUDY DESIGN | Study Aim | Medication | Sample Size N (Subgroup n) | Age at Study Entry, Mean or Median (SD or Range) in Years | Study Length *Follow Up Duration* | Grade |
|---|---|---|---|---|---|---|---|
| Lansky et al., 1983 [33] (USA) | Cross-sectional; single center "Children", ALL | Correlate urinary assay of prednisone with demographic and psychological testing | Prednisone | 31 | 7.2 (2–14) | Single time point | VL |
| Lau et al., 1998 [34] (Canada) | Prospective observational with subgroup randomization; single center "Children", ALL | Assess adherence to 6MP; subgroup randomized to AM followed by PM medication administration to determine if timing affects adherence | 6MP | 24 *(Randomized n = 8)* | 7.3 (4.6) | *Mean (SD, range): 44 (20.2, 15–94) days* | L |
| Lennard et al., 1995 [35] (UK) | Cross-sectional; multicenter "Children", ALL | Assess use of intracellular thioguanine metabolites as indicator of nonadherence | 6MP | 327 | 5 (1–15) | Single time point | VL |
| Lennard et al., 2013 [36] (UK/Ireland) | Prospective observational; multicenter 1–18 year old, ALL [d] | Assess TPMT phenotype-genotype concordance; influence of TPMT on thiopurine metabolite formation; use of metabolites as marker of nonadherence | 6MP | 1194 *(6TG n = 450; 6MP n = 744)* | NR (1–18) | Data collected over 2 years | M |
| Lennard et al., 2015 [37] (UK/Ireland) | Prospective observational; multicenter 1–18 year old, ALL [d] | Assess *TPMT* polymorphism on thiopurine dose intensity, myelosuppression and treatment outcomes; use of metabolites as marker of nonadherence | 6MP | 1082 | NR (1–18) <2 (8%); 2–9 (77%); 10–18 (15%) | Data collected over 2 years | M |
| MacDougall et al., 1992 [38] (South Africa) | Cross-sectional; single center 3–14 year old, ALL | Assess use of urine 6MP assay as indicator for adherence | 6MP | 21 | NR (3–14) | Single time point | VL |

**Table 1.** *Cont.*

| Source, Year (Country) | STUDY DESIGN | Study Aim | Medication | Sample Size N *(Subgroup n)* | Age at Study Entry, Mean or Median (SD or Range) in Years | Study Length *Follow Up Duration* | Grade |
|---|---|---|---|---|---|---|---|
| Mancini et al., 2012 [39] (France) | Cross-sectional; multicenter all ages, multiple malignancies | Assess concordance between self-reported and physician reported adherence to oral chemotherapy, and factors associated with nonadherence | 6MP, MTX | 52 *(ALL n = 49)* | 8 (3–77) Children [<11] (60%); Adolescent [11–17] (23%); Adult [>17] (17%) | Data collected over 7 months | L |
| Pai et al., 2008 [40] (USA) | Prospective observational; multicenter 12–19 year old, ALL | Assess concordance between self-reported adherence to 6MP and intracellular metabolites among adolescents | 6MP | 51 | 15 (12–19) | Data collected over 4 months | L |
| Phillips et al., 2011 [41] (UK) | Prospective single arm pilot study; multicenter "Children", ALL | Assess safety and parental satisfaction of home-based maintenance intervention to improve adherence to oral chemotherapy | 6MP, MTX | 50 | 8 (3–19) | Data collected over 2 years | M |
| Psihogios et al., 2021 [42] (USA) | Prospective observational; single center 15–25 year old, ALL | Assess feasibly and acceptability of text-based assessment of adherence to 6MP | 6MP | 18 | 17.94 (2.31) | Data collected over 28 days | L |
| Rohan et al., 2015 [43] (USA) | Prospective observational; multicenter 7–19 year old, ALL/LBL c | Assess adherence to 6MP and relationship to patient demographics | 6MP | 139 | 12.3 (3.4) | Data collected over 30 days | M |
| Rohan et al., 2017 [44] (USA) | Prospective observational; multicenter 7–19 year old, ALL/LBL c | Assess concordance of pharmacological (intracellular metabolites) and behavioral (MEMS) measures of 6MP adherence | 6MP | 139 | 12.3 (3.4) | Data collected over 15 months | M |

| Source, Year (Country) | STUDY DESIGN | Study Aim | Medication | Sample Size N (Subgroup n) | Age at Study Entry, Mean or Median (SD or Range) in Years | Study Length Follow Up Duration | Grade |
|---|---|---|---|---|---|---|---|
| Schroder et al., 1986 [45] (Denmark) | Cross-sectional; multicenter "Children", ALL | Describe pharmacokinetics of MTX in erythrocytes during maintenance therapy; assess use as marker of nonadherence | MTX | 47 | NR | Single time point | VL |
| Schroder et al., 1987 [46] (Denmark) | Cross-sectional; multicenter "Children", ALL | Describe pharmacokinetics of MTX in neutrophils during maintenance therapy; assess use as marker of nonadherence | MTX | 16 | NR | Single time point | VL |
| Smith et al., 1979 [47] (USA) | Prospective observational; single center "Children", multiple malignancies (subgroup ALL/LBL) | Assess prednisone adherence in pediatric malignancies | Prednisone | 52 (ALL n = 43) | NR (0.67–17) | Data collected over 16 months | L |
| Wu et al., 2008 [48] (USA) | Cross-sectional; national database ≤21 year old, ALL | Assess adherence to 6MP and MTX using prescription refills recorded in national claims database Medical Outcomes Research for Effectiveness and Economics (MORE) Registry | 6MP, MTX | 900 | 12.7 (4.2) Children [<12] (42%); Adolescents [12–17] (42%); Young adult [18–21] (16%) | Single time point | L |

Abbreviations: *6MP, Mercaptopurine; MTX, Methotrexate; 6TG, Thioguanine.* [a] Cohort from COG AALL03N1 study to assess compliance with long-term mercaptopurine treatment in young patients with ALL. [b] Cohort from GBTLI-93 randomized multicenter trial of 18 months vs. 24 months of maintenance therapy. [c] Cohort from 15-month randomized multicenter trial of family-centered problem-solving intervention to promote medication adherence in pediatric cancer. Randomization of family problem solving training compared to current psychosocial care did not affect adherence. Observational data of 6MP adherence measured by electronic monitoring device (Medication Event Monitoring System [MEMS]). [d] Cohort from MRC ALL97 and ALL97/99 randomized multicenter trial assessing 6MP vs. 6TG and dexamethasone vs. prednisone in maintenance therapy. Add-on pharmacogenetic and drug metabolism study to assess inter- and intra-patient variability in response to oral thiopurines.

Data from four multicenter studies were analyzed and reported in more than one article. Six articles reported analyses from the Children's Oncology Group (COG) AALL03N1 [13–15,23,31,32], a prospective observational study to assess adherence to long-term 6-MP treatment in young patients with ALL. Two articles reported on observational adherence data from a prospective randomized control trial by the Brazilian Cooperative for Treatment of Childhood ALL (GBTLI) ALL-93 comparing treatment outcomes with 18-month versus 24-month maintenance therapy durations [18,19]. Participants were analyzed as a whole, not separated by treatment arm. Three articles reported on observational adherence data from a multicenter randomized control trial comparing a family-centered, problem-solving training intervention to current psychosocial care [24,43,44]. This trial ultimately did not yield any differences in adherence; thus, adherence analyses was not separated by treatment arm. Two articles reported observational adherence data from maintenance therapy of Medical Research Council (MRC) ALL97 and ALL97/99 [36,37], a prospective randomized control trial evaluating outcome differences between treatment with dexamethasone versus prednisone, and 6MP versus 6TG. The articles separately analyzed adherence to 6MP and 6TG. Most articles received a COCHRANE grade of low (*n* = 13) due to observational studies, or very low (*n* = 8) due to downgraded observational studies for limitations in design and imprecise results.

## 4. Assessment of Adherence and Prevalence of Nonadherence

Table 2 summarizes adherence assessments of the included publications. Measures of adherence were either subjective or objective. Almost all articles included at least one objective measure of adherence; 38% included both. Only 3 publications reported exclusively subjective measures. Adherence rates were reported as frequency with which an individual patient was taking medication as prescribed. Nonadherence was calculated as a prevalence of the study population who were not taking medication as prescribed, typically based on a specific medication adherence cut off.

**Table 2.** Assessment of nonadherence and clinical outcomes.

| Author, Year | Adherence Assessment [S] Subjective [O] Objective | Adherence Rate | Definition and *Prevalence* of Nonadherence | Clinical Outcomes Related to Nonadherence |
|---|---|---|---|---|
| **Mercaptopurine** | | | | |
| Alsous et al., 2017 [12] | [S] Survey—MARS [a] for parents and adolescents [O] Metabolite—TGN, MMP [b] | NR | [S] MARS score <90%: *parents 5.8% (n = 3/52)*, **adolescents 0%** *(n = 0/15)* [O] TGN and MMP <20%ile: **15.4%** *(n = 8/52)*<br><br>Overall detected by at least 1 method: **19.2%** *(n = 10/52)* | NA |
| Bhatia et al., 2012 [13] | [O] Electronic—MEMS [c] | 94.7% month 1 to 90.2% month 6 | [O] MEMS adherence <95%: **44%** *(n = 142/327)* | **Increased incidence and risk of relapse with nonadherence** Cumulative incidence of relapse at 4 years: 11% (nonadherent 17% vs. adherent 4.9%, *p* = 0.0001); Relapse OR2.5 (*p* = 0.002); Adjusted risk of relapse attributed to nonadherence 58.8% |
| Bhatia et al., 2014 [14] | [O] Electronic—MEMS | 95% month 1 to 91.8% month 5 | [O] MEMS adherence <90%: *Overall 20.5% (n = 61/298); non-Hispanic white 13% (n = 20/159), Asian 15% (n = 11/71), African American/black 44% (n = 30/68)* | **Increased risk of relapse with nonadherence** Relapse: 6.4% (*n* = 19/298); Relapse risk from nonadherence HR3.9 (*p* = 0.01); Adjusted risk of relapse attributed to nonadherence 33% |
| Bhatia et al., 2015 [15] | [O] Electronic—MEMS | NR | [O] MEMS adherence <95%: **42%** *(n = 198/470)* | **Increased incidence and risk of relapse with nonadherence** Cumulative incidence of relapse at 6 years: 9% (nonadherent 13.9% vs. adherent 4.7% *p* = 0.001); Relapse risk from nonadherence HR2.7 (*p* = 0.01) Varying metabolite (TGN) levels not predictive of relapse overall, but among adheres, highly variable TGN levels can predict relapse (HR4.4, *p* = 0.02) |
| Bhatia et al., 2020 [16] | [O] Electronic—MEMS<br><br>* Intervention: Education + daily text reminders prompting supervised therapy | Intervention group: Baseline 92.2%; post 94% Education only group: Baseline 93.5%; post 92.5% | [O] MEMS adherence <95%: **Baseline 31%** *(n = 138/444)*<br><br>Intervention group: Baseline 32% (*n* = 74/230); post 35% (*n* = 81/230) Education only group: Baseline 29.5% (*n* = 64/214); post 41% (*n* = 88/214) | **Intervention did not improve overall prevalence of nonadherence** (*p* = 0.08), but intervention increased mean adherence rate in patients ≥12 years old (93% vs. 90%, *p* = 0.04) and ≥12 year old with baseline adherence <90% (83.4% vs. 74.6%, *p* = 0.008) |

**Table 2.** *Cont.*

| Author, Year | Adherence Assessment [S] Subjective [O] Objective | Adherence Rate | Definition and *Prevalence* of Nonadherence | Clinical Outcomes Related to Nonadherence |
|---|---|---|---|---|
| **Mercaptopurine** | | | | |
| Davies et al., 1993 [17] | [S] Interview of parents [O] Metabolite—TGN | NR | [S] Admitted nonadherence: **9%** *(n = 2/22);* Equivocal history of adherence: *27% (n = 6/22)* [O] Wide fluctuation of TGN level[b]: **27%** *(n = 6/22)* | NA |
| De Oliveira et al., 2004 [18] | [S] Interview of parents [O1] Review of medical chart [O2] Metabolite—TGN, MMP | NR | [S] Report 2+ missed doses: **33%** *(n = 13/39)* [O1] Record of interruption or irregular dose administration: **30.7%** *(n = 12/39)* [O2] Significant TGN and MMP decrease without decrease in prescribed dose: **16.6%** *(n = 6/36)* Overall detected by at least 1 method: **53.8%** *(n = 21/39);* by at least 2 methods: **20.5%** *(n = 8/39)* | **Increased relapse prevalence in nonadherent group** Relapse: 26% (*n* = 10/39); nonadherent 33% (*n* = 7/21) vs. adherent 17% (*n* = 3/18) |
| De Oliveira et al., 2005 [19] | [S] Interview of parents [O] Review of medical chart | NR | [S] Report 2+ missed doses: **27%** *(n = 20/73)* [O] Record of interruption or irregular dose administration: **30%** *(n = 22/73)* | **No difference in EFS and relapse with nonadherence** Overall 8.5 year EFS 72.4% (nonadherent 72% vs. adherent 72.8%, *p* = 0.88); Relapse: 25% (*n* = 18/73); nonadherent 25% (*n* = 5/20) vs. adherent 25% (*n* = 13/25) |
| Hawwa et al., 2009 [21] | [S] Survey—MAS-4 [d] for parents [O] Metabolite—TGN, MMP | NR | [S] MAS ≥2: **15.8%** *(n = 3/19)* [O] Low TGN and MMP cluster: **21.1%** *(n = 4/19)* [O] Wide fluctuation of TGN level: **5.3%** *(n = 1/19)* Overall detected by at least 1 method: **26.3%** *(n = 5/19)* | |
| Heneghan et al., 2020 [22] | [S1] Survey—MMAS-8 [e] for parents and adolescents [S2] Survey—VAS [f] for parents and adolescents | NR | [S1] MMAS <8: **Parents 43%** *(n = 21/49);* **adolescents 73%** *(n = 12/16)* [S2] VAS <95%: **Parents 10%** *(n = 5/49);* **adolescents 12%** *(n = 2/16)* | |

| Author, Year | Adherence Assessment [S] Subjective [O] Objective | Adherence Rate | Definition and *Prevalence* of Nonadherence | Clinical Outcomes Related to Nonadherence |
|---|---|---|---|---|
| **Mercaptopurine** | | | | |
| Hoppmann et al., 2021 [23] | [O] Electronic—MEMS | NR | [O] MEMS <95%: **36%** *(n = 148/407);* MEMS <90%: **28%** *(n = 115/407)* <br><br> Month 3 data for MEMS <90% used to develop prediction model; predicated probability of 0.3 used as cut off for binary risk classifier of high or low risk of nonadherence with sensitivity 71%, specificity 76% | **Risk of relapse higher with higher probability of nonadherence** Cumulative incidence of relapse in 5 years: 11.9% for at high-risk nonadherence vs. 4.5% for at low risk ($p = 0.006$) Relapse risk at high risk nonadherence HR2.2 ($p = 0.07$) |
| Isaac et al., 2020 [24] | [O] Electronic—MEMS | Mean (SD): Non-minority 82.5% (3.3%), minority 82.3% (1.5%), $p > 0.05$ | NR | Relapse: 8.6% (n = 12/139) |
| Kato et al., 2008 [26] | [S1] Survey—MAS-4 [d] *(n = 375)* [S2] Survey—CDCI [g] *(n = 375)* [O] Metabolite—MMP *(n = 54)* <br><br> * Intervention: Cancer based videogame | Mean MAS-4 score (SD): Intervention 2.9 (1.1), control 3.0 (1.1) ($p$ = ns) Mean CDCI score (SD): Intervention 81 (8.7), control 78.4 (7.5) ($p$ = ns) | [O] MMP level <1000 pmol/8 × $10^8$ erythrocytes: **NR**, *lower nonadherence with intervention than control, p < 0.001* | **Videogame intervention significantly improved prevalence of nonadherence** |
| Khalek et al., 2015 [27] | [S] Survey for parents [O] Drug level—serum 6MP | NR | [S] Reported 2+ missed doses: **55%** *(n = 71/129)* [O] Serum 6MP <50%ile (<9.3 ng): **50%** *(n = 65/129)* | |
| Kremeike et al., 2015 [28] | [S] Survey for parents [O] Metabolite—TGN, MMP | NR | [S] Reported non-exact medication intake: **12%** *(n = 4/33)* [O] TGN and MMP below therapeutic range: **TGN 58%** *(n = 23/40)*, **MMP 67%** *(n = 27/40)*M | |
| Kristjansdottir et al., 2021 [29] | [O] Metabolite—TGN, MMP | NR | [O] Undetectable MMP in TPMT WT: **9.8%** *(n = 5/51)*; TGN <100 nmol/mmol hemoglobin with normal ALT and wbc: **13.7%** *(n = 7/51)*; Wide fluctuation in TGN level: **52.6%** *(n = 20/38)* <br><br> Overall detected by at least 1 method: **49%** *(n = 25/51)* | **No association between nonadherence and relapse risk** Relapse: 11.3% (n = 7/62) 5-year DFS 78%, OS 91.7% |
| Lancaster et al., 1997 [30] | [O] Metabolite—TGN | NR | [O] Undetectable TGN level: **2%** *(n = 9/496)* | |

**Table 2.** *Cont.*

| Author, Year | Adherence Assessment [S] Subjective [O] Objective | Adherence Rate | Definition and *Prevalence* of Nonadherence | Clinical Outcomes Related to Nonadherence |
|---|---|---|---|---|
| **Mercaptopurine** | | | | |
| Landier et al., 2017 [31] | [S] Survey for parents and adolescents [O] Electronic—MEMS | Self-report 92.6% MEMS 91.0% | [O] MEMS <95%: *39.7% (n = 165/416)* <br><br> Perfect reporter (MEMS matched self-report): *12% (n = 50/416);* Over reporter (Self-report > MEMS): *23.6% (n = 98/416)* | **Self-report overestimates intake, especially in nonadherent patients.** 88% (*n* = 366/416) had self-report > MEMS at least some of the time. Nonadherent patients were more likely (OR9.4) to overestimate intake. Self-report sensitivity 52.7%, specificity 95.8% for detecting nonadherence |
| Landier et al., 2017 [32] | [O] Electronic—MEMS | MEMS 91% | [O] MEMS <95%: *48.3% (n = 193/441)* | **No association between relapse risk and ingestion habits** Cumulative incidence of relapse at 5 years: 8.6%. No difference in taking with food, with dairy, in morning or evening |
| Lau et al., 1998 [34] | [O] Electronic—MEMS | NR | [O] MEMS <95%: *58% (n = 14/24);* MEMS <90%: *33% (n = 8/24)* | |
| Lennard et al., 1995 [35] | [O] Metabolite—TGN, MMP | NR | [O] TGN and MMP <25%ile: *10% (n = 32/237)* | |
| Lennard et al., 2013 [36] | [O] Metabolite—TGN, MMP | NR | [O] Undetectable TGN and MMP: *2.7% (n = 20/744)* | |
| Lennard et al., 2015 [37] | [O] Metabolite—TGN, MMP | NR | [O] Undetectable TGN and MMP: *2.8% (n = 20/707);* TGN and MMP <25%ile: *10% (n = 71/707)* | 5 year EFS 80%, OS 89%. **No difference in EFS with nonadherence to 6MP.** |
| MacDougall et al., 1992 [38] | [O] Drug level—urine 6MP | NR | Unable to detect nonadherence due to variability in 6MP urine excretion and unpredictable pattern of night-time voids Prevalence of adherence [O] Detectable urine 6MP in first morning voids of PM 6MP takers: *81% (n = 17/21)* | |

| Author, Year | Adherence Assessment [S] Subjective [O] Objective | Adherence Rate | Definition and *Prevalence* of Nonadherence | Clinical Outcomes Related to Nonadherence |
|---|---|---|---|---|
| **Mercaptopurine** | | | | |
| Pai et al., 2008 [40] | [S] Interview for patients [O] Metabolites—TGN, MMP | NR | [S] Reported missed dose in past week: **24.5%** *(n = 14/51)*; Missed dose in past 2 weeks: **45.1%** *(n = 23/51)* [O] TGN and MMP <95%ile: **52.9%** *(n = 27/51)* | **Self-report at month 2 predicts nonadherence at month 4** (OR3.54, *p* < 0.05) |
| Psihogios et al., 2021 [42] | [S1] Survey—Text survey for patients (*n* = 18) [S2] Survey for physicians (*n* = 16) [O] Electronic—MEMS (*n* = 15) <br><br> * Intervention: Text survey to assess adherence | Text 96.8%; Physician 97.8%; MEMS 90.7% <br><br> # missed doses mean (SD): Text 0.89 (1.64); Provider 0.63 (0.96); MEMS 2.6 (3.09) | NR | Daily text messages feasible and reliable for delivering medication adherence assessment |
| Rohan et al., 2015 [43] | [O] Electronic—MEMS | Baseline 86.2%, decline to 83% in 1 month | [O] MEMS <95%: **44%** *(n = 58/139)*; MEMS <90%: **35%** *(n = 46/139)* | |
| Rohan et al., 2017 [44] | [O1] Metabolite—TGN, MMP [O2] Electronic—MEMS | MEMS—low TGN/low MMP: 72–78%; Low TGN/high MMP 85–90%; high TGN/low MMP 86–89% (*p* = 0.008) | [O1] Low TGN and MMP cluster: **40.8%** *(n = 312/764)* [O2] MEMS <95%: ***Low TGN/low MMP group (nonadherent metabolite) 60.3–74.2%; Low TGN/high MMP (adherent metabolite) 42.4–56.4%*** | |
| Wu et al., 2008 [48] | [O] Review of prescription claims | Medication possession ratio [i] **6MP 85%** | NR | |
| **Methotrexate** | | | | |
| Jaime-Perez et al., 2009 [25] | [S] Interview for parents [O1] Review of medical charts [O2] Drug level—serum MTX | NR | [S] Reported 2+ missed doses: **10%** *(n = 5/49)* [O1] Record of 2+ missed doses: **16.3%** *(n = 8/49)* [O2] Undetectable serum MTX level: **29%** *(n = 14/49)* | |
| Kremeike et al., 2015 [28] | [S] Survey for parents | NR | [S] Reported non-exact medication intake: **MTX 33%** *(n = 7/31)* | |

**Table 2.** *Cont.*

| Author, Year | Adherence Assessment [S] Subjective [O] Objective | Adherence Rate | Definition and *Prevalence* of Nonadherence | Clinical Outcomes Related to Nonadherence |
|---|---|---|---|---|
| **Mercaptopurine** | | | | |
| Schroder et al., 1986 [45] | [S] Interview for parents [O] Drug level—erythrocyte MTX | NR | [O] Undetectable eMTX level: **6%** *(n = 3/47)* [S] Admitted nonadherence: **4%** *(n = 2/47)* | |
| Schroder et al., 1987 [46] | [S] Interview for parents [O] Drug level—neutrophil MTX level | NR | [O] Undetectable nMTX level: **5%** *(n = 1/19)* [S] Admitted nonadherence: **5%** *(n = 1/19)* | |
| Wu et al., 2008 [48] | [O] Review of prescription claims | Medication possession ratio [i] MTX 81% | Did not provide information about prevalence of nonadherence | |
| Prednisone | | | | |
| Lansky et al., 1983 [33] | [O] Metabolite—urine 17 kgs/Cr | NR | [O] Average urine 17-kgs/Cr value <18.7: **42%** *(n = 13/31)* | |
| Smith et al., 1979 [47] | [O] Metabolite—urine 17-kgs/Cr | NR | [O] Average urine 17-kgs/Cr <18.7: **33%** *(n = 9/27)* | |
| Not specified | | | | |
| Farberman et al., 2021 [20] | [S] Survey—SMAQ [h] for parents, adolescents, physicians | NR | [S] SMAQ nonadherent: ***parents 25%*** *(n = 48/194);* ***adolescent 55%*** *(n = 20/37);* ***physician 18%*** *(n = 37/203)* | |
| Mancini et al., 2012 [39] | [S1] Survey (3 questions) and interview for parents and adolescents [S2] Survey for physicians | NR | [S1] MMAS-3 >1 or reported 1+ missed doses in past week: ***Overall 23%*** *(n = 12/52); children 13% (n = 4/31); adolescents 33% (n = 4/12); adults 44% (n = 4/9)* [S2] Physician reported missed dose: **11.5%** *(n = 6/52)* | |

**Table 2.** *Cont.*

| Author, Year | Adherence Assessment [S] Subjective [O] Objective | Adherence Rate | Definition and *Prevalence* of Nonadherence | Clinical Outcomes Related to Nonadherence |
|---|---|---|---|---|
| **Mercaptopurine** | | | | |
| Phillips et al., 2011 [41] | [O] Tablet count<br><br>* Intervention: Home-based maintenance program | | [O] Tablet count <97% adherence: *After* **3 months intervention 72%** *(n = 35/50);* **After remediation 22%** *(n = 11/50);* **After 2 years 45%** *(n = 23/50)*<br><br>* Remediation: program wide education, specific confrontation with parental intervention and directly observed medication therapy | |

Abbreviations: *6MP, Mercaptopurine; MTX, Methotrexate; 6TG, Thioguanine*; CDCI, Chronic Disease Compliance Instrument; MARS, Medication Adherence Report Scale; MAS-4, Morisky Adherence Scale 4 item; MMAS-3, Modified Morisky Adherence Scale 3-item; MMAS-8, Modified Morisky Adherence Scale 8-item; MMP, methylmercaptopurine; MEMS, Medication Event Monitoring System; SMAQ, Simplified Medication Adherence Questionnaire; TGN, thioguanine nucleotides; VAS, Visual Analogue Scale; Urine 17 kgs/Cr, ratio of urine 17-ketogenic steroids/urine creatine. [a] Medication Adherence Report Scale (MARS) is a self-report adherence questionnaire with score 0–5; higher scores indicate better adherence; score of 4.5/5 is 90%. [b] Thiopurine metabolites thioguanine nucleotides (TGN) and methylmercaptopruine (MMP) are intracellular metabolites measured in erythrocytes. In hierarchical cluster analysis, low TGN/low MMP is indicative of nonadherence due to inability to explain on metabolic grounds. Wide fluctuation of TGN is defined as ratio of highest TGN level to lowest TGN level $\geq$1.9. [c] Medication Event Monitoring System (MEMS) adherence is calculated as ratio of # days with MEMS cap opening to # days of medication prescribed as a percent. [d] Morisky Adherence Scale 4-item (MAS-4) is a self-report adherence questionnaire with 4 dichotomous questions with score 0–4; higher scores indicate poorer adherence. [e] Modified Morisky Adherence Score 8-item (MMAS-8) is a self-report adherence questionnaire with 7 yes/no questions and 1 multiple-choice with score 0–8; higher score indicates better adherence, dichotomized into adherent score 8 vs. nonadherent <8. Evaluates adherence over past 2 weeks. [f] Visual Analogue Scale (VAS) is a self-report adherence measure, estimates along a continuum % 6MP doses taken as prescribed over the last month. Adherent is taking $\geq$95% of doses as prescribed, nonadherence is taking <95% doses as prescribed. [g] Chronic Disease Compliance Instrument (CDCI) is a self-report adherence questionnaire for adolescents with cancer composed of 18 questions with score 18–90; higher scores indicate better adherence. [h] Simplified Medication Adherence Questionnaire (SMAQ) is a self-report adherence questionnaire with 6 questions. Dichotomized into adherent and nonadherent. Nonadherent defined as positive response to any qualitative question, more than 2 doses missed in the past week, or over 2 days of total non-medication during the past 3 months. [i] Medication possession ratio calculated as [sum of the number of days of the medication supplied]/[days in maintenance phase] expressed as percent.

## 5. Pharmacologic Adherence

### 5.1. Thiopurines (6-mercaptopurine)

Daily thiopurines, predominantly 6-mercaptopurine (6MP), make up the backbone of maintenance therapy for acute leukemias. In our review, 15 of the 28 publications analyzing adherence to thiopurines reported pharmacological measurements of 6MP. Measurements of the thiopurine metabolites (thioguanine nucleotides [TGN] and methylmercaptopurine [MMP]) were the most reported (*n* = 13) [12,17,18,21,26,28–30,35–37,43,45]. Direct measures of serum 6MP level [27] and urine 6MP level [38] were reported in one article each. Overall, prevalence of nonadherence to 6MP using pharmacological assessment was widely variable 2–67%. There was no consensus definition of nonadherence. The most common definition of nonadherence was based on low MMP and TGN levels (*n* = 7) which reflected chronic low exposure. However, the cut off values for low were not consistent. Prevalence of overt nonadherence using undetectable metabolite levels was the lowest at 2–3%. Prevalence of partial nonadherence was detected using various definitions, including: metabolite level cut offs in the lower quartile, or less than 20th percentile (10–15%); relative decrease in metabolite level without decrease in medication dose (17%); hierarchical cluster analysis to determine low TGN and low MMP group (21–53%); single metabolite level below therapeutic range (58–67%). Nonadherence was also defined using variability in TGN levels, with prevalence of 5–27%. Use of direct measurements of mercaptopurine, whether in serum or urine, was limited (*n* = 2) due to variable inter-patient pharmacokinetics and rapid drug metabolism.

### 5.2. Methotrexate

Weekly low-dose methotrexate (MTX) is an integral component of maintenance therapy for ALL. Thus, 3 of the 5 publications reporting prevalence of nonadherence to oral MTX used pharmacological measures, including serum MTX [25], erythrocyte MTX (eMTX) [45], and neutrophil MTX (nMTX) [46]. Like serum measurements of 6MP, MTX levels in serum drop rapidly after drug ingestion and metabolites accumulate intracellularly in erythrocytes and neutrophils. Overall, prevalence of nonadherence to MTX using pharmacological assessment was between 5–29%. All three methods used undetectable level as the definition of nonadherence. Undetectable serum MTX yielded a higher prevalence of nonadherence of 29%, compared to undetectable measurements of intracellular MTX metabolites (eMTX or nMTX) yielding prevalence of 5–6%.

### 5.3. Steroids

Intermittent pulses of steroids, such as prednisone, are a major component of maintenance therapy. Two publications evaluated adherence to prednisone during maintenance therapy for ALL [33,47]. Both were over 30 years old and used urine metabolites as a measure of nonadherence. Prevalence rates of nonadherence reported in these 2 articles were 33–42%. Smith et al. [47] also evaluated hemoglobin rise and weight gain as objective measures of adherence to prednisone, and did not demonstrate any significant relationship.

## 6. Behavioral Adherence

### 6.1. Medication Event Monitoring System (MEMS)

The MEMS cap is an objective, indirect measure of adherence to therapy. Microelectronic technology is used to record date and time of each pill bottle opening but does not provide information about ingestion of the medication [13]. Herein, 12 of 37 (32%) articles used MEMS as an assessment of adherence [13–16,23,24,31,32,34,42–44]. Adherence was calculated as ratio of number of days with MEMS cap opening to number of medications prescribed as a percent. MEMS adherence over the course of 1 month ranged from 82–95%. Nonadherence was defined as MEMS adherence <90% or <95% based on an association with statistically significant increase in relapse when MEMS adherence rates fell below 90 or 95% [13]. All 12 publications assessed adherence to 6MP, with 9 reporting prevalence of nonadherence. In all except one [34], parents/patients were informed about the purpose

of the MEMS and were instructed to take all doses of 6MP from the MEMS bottle. The prevalence of nonadherence for 6MP detected by MEMS was 21–58%. This was similar to the prevalence defined by low TGN and MMP metabolite cluster (21–41%) or TGN and MMP <95th percentile (53%).

Several studies sought to evaluate the relationship between MEMS behavioral adherence and pharmacological adherence measures [13,14,44]. Bhatia et al. [13,14] showed higher MEMS adherence correlating with increased TGN levels after accounting for *TPMT* activity and 6MP dose intensity. Rohan et al. [44] demonstrated that pharmacological nonadherence (defined by low TGN and MMP cluster) was associated with higher prevalence of behavioral nonadherence, defined by MEMS adherence <95%—63% compared to 48% seen in the low TGN and high MMP cluster. Furthermore, Bhatia et al. [15], in a third publication, showed a correlation between high intra-individual variability in MEMS adherence (coefficient of variation ≥85th percentile) and nonadherence, which was consistent with Davies et al. [17], demonstrating association of widely fluctuating TGN levels with nonadherence.

### 6.2. Tablet Counting

Tablet counting is an economical, objective, indirect measure of adherence which requires patients/families to return the number of pills prior to next prescription. Investigators can determine how many pills should be returned based on the month's prescription and any dose adjustments made during the month [49]. Like electronic monitoring, it does not provide information about ingestion of the medication. One study used tablet counting to evaluate adherence to daily 6MP and weekly MTX [41]. Nonadherence was defined as a tablet count difference greater than 3% from the prescribed number, implying an adherence of less than 97%. Prevalence of nonadherence during the baseline 3 months was 72%, higher than prevalence reported using other objective measures included in our review. Participants received remediation measures and, in the subsequent 3 months, prevalence of nonadherence decreased to 22%, but increased to 45% at 2 years. Unlike the studies with MEMS cap, there have been no studies to correlate pharmacological measures indicating increased medication exposure with tablet count adherence.

### 6.3. Prescription Review

Medication refill records can provide an objective, indirect measure of adherence of a large population of patients. Unlike electronic monitoring or tablet counting, refill records can minimize potential for patient or parent reactivity to being monitored [49]. However, refill records alone do not account for dose adjustments or provide information about medication ingestion. Only one publication used prescription review to assess adherence to daily 6MP and weekly oral MTX [48]. This study used a national claims database (Medical Outcomes Research for Effectiveness and Economics Registry) with all inpatient and outpatient claims and dispensed prescription medication claims. Adherence was measured as medication possession ratio (MPR), defined as sum number of days of medication supplied/days in maintenance phase. Median MPR to 6MP was 85%, MTX was 81%. There was no defined parameter for nonadherence and no prevalence of nonadherence reported.

### 6.4. Medical Chart Review

Several articles used review of medical charts as a measure of nonadherence, using records of interruption or irregular dose administration [18,19,25]. This method detected a nonadherence prevalence of 30–31% for 6MP and 16% for MTX.

## 7. Subjective Adherence
### 7.1. Self-Report

Seventeen of 37 (46%) articles used self-reporting (participant survey or interview) as a subjective measure of adherence. Various methods were used, including structured or semi-structured interviews or questionnaires of patient and/or parents, specific validated

questionnaires (Modified Morisky Adherence scale (MMAS) [3-, 4-, 8-item], Medication Adherence Report Scale, Chronic Disease Compliance Instrument, Visual Analogue Scale (VAS), Simplified Medication Adherence Questionnaire), and text survey. Self-report measures asked about missed medication doses in the past 1–2 weeks [39,40] or over the entire course of maintenance therapy [18,19,25,27,28]. The majority of the interviews and questionnaires were directed toward parents (*n* = 10); several included adolescents and young adults greater than 11 years old (*n* = 5). Self-reported medication adherence rate via surveys was between 93–97%. Definition of nonadherence was variable depending on method of self-reporting, listed in Table 2. Prevalence of nonadherence using these definitions were widely variable, between 0–73% for daily 6MP and 10–33% for weekly MTX. There was no consensus definition of nonadherence in self-reporting. Although each of the validated questionnaires had its own definition of nonadherence based on specific score cut offs, none of the publications used the same questionnaire, making comparisons between publications difficult. For example, Alsous et al. [12] used the Medication Adherence Report Scale (MARS) with score 0–5 (higher scores indicating better adherence) and a score of 4.5/5 (90%); they found prevalence of nonadherence of 5.8% based on parental response and 0% based on adolescent response. Even within the same study population, Heneghan et al. [22] found different rates of nonadherence using 2 different validated questionnaires—Modified Morisky Adherence Score 8-item (MMAS-8) and the Visual Analogue Scale. MMAS responses showed a prevalence of nonadherence of 43% based on parental responses and 73% based on adolescent responses, and VAS responses indicated a prevalence of nonadherence of 10% based on parental response and 12% based on adolescent response.

Survey or interview responses were also variable in their estimates of nonadherence depending on the definition of nonadherence and period of recall. When nonadherence was defined based on reports of not taking the medication, the prevalence was consistently low: 4–5% for MTX and 9% for 6MP, similar to the prevalence of 2–9% based on undetectable intracellular drug metabolite levels of 6MP or MTX. When surveys or interviews asked about recall of missed doses (at least 2) or non-exact medication administration over the entirety of maintenance, the prevalence of nonadherence increased, to 12–55% for 6MP and 10% for MTX. When the recall period was shortened to the 1–2 weeks immediately prior, the prevalence of nonadherence, defined as missing 1 or more doses, was 25% for 6MP and 23% for MTX within 1 week, and 45% for 6MP within 2 weeks.

Four publications included both parent and adolescent responses to surveys. Adolescent responses resulted in 20–30% higher prevalence of nonadherence compared to parent responses [20,22,39]. Alsous et al. [12], on the other hand, detected a lower prevalence of nonadherence based on adolescent response compared to parental response (0% and 5.8% respectively), though both were significantly lower than prevalence reported in the other 2 publications. Heneghan et al. [22] analyzed 7 parent–adolescent dyads included in their sample and found no significant correlation between parent and adolescent responses on either self-report measure (VAS or MMAS-8), suggesting parents may not have provided accurate estimates of adolescent medication adherence.

### 7.2. Text Messaging

Psihogios et al. [42] performed a pilot study evaluating the use of daily text messages to assess 6MP adherence in adolescent and young adult patients. They found no significant correlation between text message response of adherence (97%) with MEMS cap opening (97%) over a 28-day period. However, the data did converge on a majority (>90%) of days and the authors concluded that daily text messages could be an acceptable and feasible method of assessing medication adherence.

### 7.3. Provider Survey

Three publications included surveys of physicians as a subjective measure of patients' medication adherence. Farberman et al. [20] and Mancini et al. [39] showed lower

prevalence of nonadherence detected by physician survey responses compared to parent/adolescent responses (12–18% and 25–55%, respectively). However, Psihogios et al. [42] demonstrated similar adherence rates between participant text survey responses and physician survey responses (97% and 98%, respectively).

## 8. Comparison of Objective and Subjective Measures

Self-report measures tend to over-estimate adherence and are not very reliable on their own [49]. Fourteen publications included both an objective and subjective measure of adherence (Table 2). The majority of these publications demonstrated higher nonadherence prevalence (5–40% higher) or lower medication adherence rates (91% vs. 97%) when using objective measures, compared to self-reported results. The prevalence rates of nonadherence were similar or lower compared to self-reporting when using the following objective adherence assessments: medical chart review [18,19], undetectable serum 6MP level [27], relative decrease in 6MP metabolite level without medication dose adjustments [18], and widely fluctuating TGN levels [21]. However, when comparing data specifically to admitted nonadherence, widely fluctuating TGN yielded a 10% higher prevalence of nonadherence [17], while undetectable intracellular MTX metabolites yielded a similarly low prevalence of nonadherence [45,46]. Hawwa et al. [21] showed a 67% concordance between the MAS self-report questionnaire and 6MP metabolite clusters in detecting nonadherence. Khalek et al. [27] also showed a correlation between self-report survey and serum 6MP level. Pai et al. [40] demonstrated that self-reported nonadherence could predict later 6MP metabolite nonadherent cluster assignments, wherein patients who had lower self-reported adherence were 3.5 times more likely to be in the low MMP/low TGN cluster. On the other hand, Kato et al. [26] did not see any relationship between self-report questionnaires (MAS-4 and CDCI) and 6MP metabolite levels (MMP). Landier et al. [31] showed that 88% of patients inaccurately reported their 6MP intake (measured via MEMS), consistent with prior studies [49] demonstrating that self-reported results overestimated medication adherence. This was particularly prominent in patients in the nonadherent group (MEMS <95%), who were 9.4 times more likely to over-report 6MP intake compared to adherent patients (MEMS >95%). Over-reporting, in this case, was defined as 5 or more days in more than 50% of studied months. Furthermore, they showed that self-report results had low sensitivity (53%) but high specificity (96%), meaning patients who did not report nonadherence could go unrecognized but patients who did report were generally not taking 6MP [31]. This was consistent with what we observed in this review, i.e., a similar prevalence of nonadherence (both <10%) detected by admitted nonadherence and undetectable intracellular 6MP or MTX metabolites.

## 9. Correlates of Nonadherence

### 9.1. Age

Across various chronic illnesses, including cancer, adolescents and young adults have been shown to consistently have increased medication nonadherence [50–53]. We sought to clarify the evidence for this association between age and medication adherence in patients on oral maintenance therapy for treatment of ALL, specifically including the adolescent and young adult population, which we defined as 11–39 years old. The majority of publications had an age limit of either 18 or 21 years old, with a median or mean age of less than 11 years old (Table 1). There were only 8 publications with median or mean age greater than 11 years old with a maximum mean of 24.2 years old; five of these included patients older than 21 years old. Comparing the prevalence of nonadherence between the publications with mean or median age >11 with the remainder of publications was difficult due to the various adherence measures and definitions of nonadherence. However, in general, the reported medication adherence rates appeared lower in the publications with age >11 years (81–86%) compared to <11 years (>90%). Eleven publications commented on the relationship between age and medication adherence rate or prevalence of nonadherence. The majority of these publications noted increasing age to be associated with

increased nonadherence [13,17,30,39,47,48]. Interestingly, Phillips et al. [41], which used tablet count as an adherence measure, showed the inverse relationship, e.g., older patients were noted to be more adherent to oral maintenance medications. Additionally, there were 3 publications which showed no relationship between age and adherence [12,18,34]. Further characterization of these publications revealed that those that showed increased prevalence of nonadherence or decrease medication adherence rates generally had larger sample sizes ($n > 300$ in 3 of the 5 studies) and specifically evaluated adolescents and young adults, as a group, compared to younger children. In contrast, studies that showed no association between age and prevalence of nonadherence or medication adherence rates were all relatively small studies ($n < 80$) and did not specifically look at adolescents as a group.

*9.2. Sex*

Sex (male or female) was broken down in 29 of the 37 articles included in this review, with males reported at 35–78%. Several publications examined the relationship between sex adherence and found no association [12,18,21,23,43].

*9.3. Race/Ethnicity*

Participants' ethnicity was reported in 15 of the 37 articles. All were from the USA or Canada. Most commonly, the groups were non-Hispanic white ($n = 15$), African American/Black ($n = 11$), Hispanic ($n = 14$), and Asian ($n = 11$). Other reported groups included non-Hispanic minority ($n = 3$), Native American ($n = 1$), and others ($n = 1$). Multiple publications demonstrated increased prevalence of nonadherence in Hispanics [13,23,48], African Americans [14,23,48], and Asians [14,48]. Furthermore, these ethnicities were also associated with being over-reporters of individual medication adherence [31]. On the contrary, Rohan et al. [43] and Isaac et al. [24]—both publications having analyzed the same family problem-solving training participation intervention—demonstrated lack of association between Hispanic ethnicity and medication adherence [43] and similar adherence rates between minority and non-minority participants [24].

*9.4. Family Structure and Parental Characteristics*

In a 2019 metanalysis, Psihogios et al. [54] found family function to have an effect on medical adherence across various pediatric chronic health conditions. In our review, specific to adherence to maintenance oral chemotherapy in ALL, the role of family structure and parental education on adherence appeared to be more equivocal. Some publications demonstrated medication nonadherence to be associated with single-mother households [13], households without mothers as primary care givers [14], single-parent households with multiple children [23], large family size [27], and parents in unstable partnerships [41]. On the contrary, other publications showed no relationship between nonadherence and mothers as fulltime caregivers [23], person responsible for administration of medication [18,25], number of children under caregiver [20], family size [18,19], or family structure [43]. Furthermore, the majority of publications showed a lack of association between nonadherence to 6MP and MTX with parental education [12,18–20,25,28,43]. However, Khalek et al. [27] showed that, in Egyptian patients, low parental education was associated with nonadherence. Bhatia at al [14] found that, specifically for African American patients, low maternal education was associated with nonadherence. Moreover, Landier et al. [31] found that low parental education was a predictor of medication adherence over-reporters. All these publications, except Rohan et al. [43], focused primarily on children (mean or median age <11 years old) rather than adolescent or young adult populations. Lanksy et al. [33] found that medication adherence was more related to parental psychology than child; therefore, examining parental characteristics and family structure could be an important component to understanding factors contributing to adherence in children. However, limited information was found on the role of family structure and parental characteristics on adolescent and young adult patients. Interestingly, the majority of publications showing a relationship

between aspects of family structure or parental education and medication nonadherence were conducted in high-income countries (USA/Canada), whereas most of the publications showing no relationship were from low- or middle-income countries. This suggested that family structure and parental characteristics could be a more important contributing factor to nonadherence in high-income countries than in low- or middle-income countries.

*9.5. Socioeconomic Status*

Data evaluating the relationship between socioeconomic status and adherence were mixed. Low household income in Asian Americans [14], low socioeconomic status overall [27,39], undernourishment [18], and lack of financial solvency [20] were associated with increased nonadherence. However, multiple other publications demonstrated no correlation between household income or parental occupation and nonadherence [12,18–20,25,28,43]. Interestingly, enrollment in Children's Health Insurance Plan (CHIP), a low-cost health coverage available to children in families who earn too much to qualify for Medicaid, was associated with decreased adherence, compared to Medicaid or commercial insurance [48]. Whether or not an association existed between socioeconomic status and adherence did not seem to differ between high income and middle-income countries with 50% of publications in each group coming from middle-income countries.

*9.6. Therapy Related Factors*

The impact of 6MP ingestion timing and method of ingestion was evaluated in several publications. Landier et al. [32] demonstrated that the dose intensity-adjusted TGN levels were comparable between taking 6MP with or without food, with or without dairy, swallowed whole or crushed, and in the morning or evening, suggesting that the timing of 6MP ingestion and how it as ingested alone did not affect drug absorption. Lau et al. [34] performed a randomization as part of their study, with a small number of patients comparing 6MP ingestion in the morning to the evening, and found no statistically significant difference in adherence between the two groups. However, other studies demonstrated increased nonadherence when 6MP was taken near bedtime or not at a consistent time every day [23,32,39], but improved adherence if taken in the morning or midday [32]. Additionally, adolescents and young adult patients, specifically, were more likely to miss 6MP on the weekends (OR2.33) [42]. Despite dairy co-ingestion not affecting TGN levels, there was a possible increased risk of nonadherence if 6MP was co-ingested with diary [32]. Moreover, increased time between 2 doses of 6MP, increased consecutive 24 h without taking a dose, and varying TGN levels over time were all associated with increased nonadherence [15,43]. Given the length of maintenance therapy, several studies also noted increased nonadherence over time [13,25,41,43], though other studies found no correlation [12,34].

*9.7. Reasons for Nonadherence*

Several publications sought to understand the reasons for nonadherence. Lansky et al. [33] found that adherence was more related to parental psychology than child, though they only included children 2–14 years old in their study, with a median age of 7 years old. The most common reasons given for nonadherence were forgetfulness [19,21,25,27,39], carelessness [21], child refusal/tantrums [19,27], and low satisfaction with drug formulation and taste [28]. On the other hand, patients who were well informed about the medications and reported medications as easy to take had increased adherence [28]. Mancini et al. [39] specifically examined reasons given by adolescents and adults and found forgetfulness and discouragement in adolescents were associated with nonadherence, and hepatic side effects were associated with nonadherence in adult patients in the form of complete discontinuation of treatment. Psihogios et al. [42] found that adolescents and young adults were more likely to miss 6MP if their adherence motivation or negative affects worsened from their own typical functioning, whereas the odds of missing 6MP decreased on days on which they were able to better open communication with their parents. When looking at patient psychiatric characteristics, only

anxiety in girls correlated with improved adherence; no other psychiatric characteristics in patients were associated with adherence [33].

## 10. Survival and Outcomes

Table 2 summarizes survival and health outcomes. The goal of prolonged maintenance therapy is to prevent relapse and improve survival. Multiple publications established an association between nonadherence, assessed using several methods, and increased relapse [13–15,18,23]. Nonadherence to daily 6MP (assessed by MEMS adherence <90% or 95%) was associated with a 2.7–3.9 fold increased risk of relapse, with adjusted risk of relapse attributable to nonadherence at 33–58.8% and cumulative incidence of relapse among non-adherers at 13.9–17%, compared to adherers at 4.7–4.9% over 4–6 years [13–15]. Using highly fluctuating TGN levels as a definition of nonadherence was not predictive of relapse overall [29]. However, in patients with behavioral adherence (MEMS adherence >95%), high intra-individual variability in TGN levels was associated with a 4.4-fold increase in risk of relapse [15]. Hoppmann et al. [23] developed a risk prediction model for 6MP nonadherence using MEMS <90% as definition of nonadherence. Using adherence status from month 3, they developed a binary risk classifier to classify patients as having high or low risk of nonadherence (sensitivity 71%, specificity 76%). They found that the risk of relapse was higher for those at high risk of nonadherence, compared to those at low risk with HR2.2. Briefly, they demonstrated a cumulative incidence of relapse at 5 years of 11.9% for those at high risk of nonadherence, compared to 4.5% for those at low risk. Using self-reporting measures alone, de Olivera et al. [19] showed no difference in 8.5-year event free survival (EFS) of 72% and prevalence of relapse of 25% between adherent and nonadherent patients. However, an earlier publication by the same group [18], using both self-report and 6MP metabolite levels, demonstrated an increased prevalence of relapse in nonadherent patients—33% compared to adherent patients at 17%. Interestingly, Kristjansdottir et al. [29] and Lennard et al. [37], both studies having used thiopurine metabolites with undetectable TGN and MMP as definitions of nonadherence, showed no association with relapse [29] and EFS [37].

## 11. Interventions

Four studies included in this review attempted interventions to improve medication adherence [26,41,43,44]. Kato et al. [26] performed a randomized control trial using a cancer-specific video game intervention vs. a standard video game, to improve medication adherence, specifically in adolescents and young adults with cancer. Participants were asked to play video games 1 h a week over a 3-month period. However, adherence to the intervention itself was only 28%, and attrition rate over the 3-month period was low (17% in the intervention, 21% in the control group). Although the sample included multiple malignancies, ALL was the most common diagnosis ($n$ = 150/375). Validated questionnaires administered to participants showed no increase in self-efficacy or knowledge with the intervention. Objective measures of adherence to 6MP via metabolite levels were evaluated for the subgroup of patients on these medications. Despite low compliance to the intervention, there was significantly lower prevalence of nonadherence, as detected by MMP level.

Phillips et al. [41] used a home-based maintenance program to help improve adherence to 6-MP and PO MTX. Tablet count was used as the measure of adherence. As part of this intervention, patients had blood tests near their home/school. Chemotherapy doses were prescribed by pharmacists or specialist nurses, and doses were confirmed over the phone. Interestingly, 3 months after the initiation of the intervention, the prevalence of nonadherence was high, at 72%. Initiation of remediation measures, including educational reminders of importance of therapy, letters reinforcing message to care takers, and direct confrontation of patients and parents, improved nonadherence to 22% after 3 months, with long-term nonadherence after 2 years at 45%. The study had no control group; however, the rate of nonadherence at 2 years was similar to the rates of nonadherence noted in other

publications in this review. As such, there may have been no improvement in increasing adherence. However, 95% of families preferred the new system, as it decreased time away from home and work, missed school days, and the need for additional childcare.

Rohan et al. [43,44] performed a randomized control trial over 15 months, using family problem solving training interventions compared to current psychosocial care. The goal of the study was to test the efficacy of family-centered interventions used to address specific barriers to medication adherence, to enhance adolescent–parent communication, and to employ problem-solving strategies using behavioral reinforcement. MEMS cap and 6MP metabolites were used to measure adherence. The intervention yielded no difference in either measure of adherence.

Lastly, Bhatia et al. [16] performed a randomized control trial comparing the efficacy of a multicomponent intervention—text reminder, direct supervision of medication ingestion, and education—versus education alone on increasing adherence to 6MP. The 16-week intervention included daily personalized text reminders to parents (and patients 12 or older) to prompt supervised (by parents) ingestion of 6MP. The intervention and control groups both received the same educational video program. MEMS cap was used to measure adherence, with MEMS adherence <95% defining nonadherence. Overall prevalence of nonadherence did not change with the intervention; however, among patients 12 or older, the intervention significantly increased mean adherence rates (93% with intervention vs. 90% with control), especially in patients 12 or older who had baseline MEMS adherence <90% (83.4% with intervention vs. 74.6% with control).

## 12. Discussion

Nonadherence to oral maintenance chemotherapy is associated with poor outcomes, including increased risk of relapse in ALL [13–15]. In this systematic review, we identified 37 peer-reviewed publications that met our inclusion criteria. Most were low- to moderate quality due to methodologic limitations and/or imprecision of results. We found an overall lack of consensus in methods of assessing adherence and definitions of nonadherence. This was reflected in the wide range in prevalence of nonadherence to oral maintenance chemotherapy (0–77%) observed in the pediatric, adolescent, and young adult populations with ALL. The most prominent correlates of nonadherence were increasing age and minority race/ethnicity, with the most common reason for nonadherence being forgetfulness.

Our results suggested that MEMS caps offered the most evidence in support of a definition of nonadherence, i.e., using the MEMS cap adherence of <95% as the cutoff, due to its association with increased risk of relapse. This was not the case for metabolite or drug level cut offs, which were widely variable. Other behavioral adherence measures lacked data to support a specific definition of nonadherence. Unfortunately, the cost of MEMS caps could be prohibitive for use outside of clinical studies. A single MEMS cap cost USD $130 in 2015, with additional costs including a MEMS cap reader to download data (USD $122), software (USD $473), and bottles to ensure fit of the cap (USD $5 per piece) [55]. Additional logistical barriers could include forgetting to use the cap, losing the cap, or forgetting to bring the bottle and cap to a clinic to download data [55].

Although self-report methods are easy and low-cost, their use is complicated by substantial variations in phrasing of questions, recall time intervals, format of response scales, and modes of administration [56]. Similarly, there was significantly variability in the definition of nonadherence depending on survey or questionnaire type, leading to a widely variable prevalence of nonadherence (between 0–73%). We found self-reported oral maintenance chemotherapy adherence was >90% in all studies that reported an adherence rate, regardless of interval of recall, ranging from daily texts to questionnaires asking about adherence over the entire course of maintenance. These measures may be further influenced by social desirability and recall accuracy [57] impacting reporting of medication adherence. Studies correlating MEMS cap data with self-reporting in adults with hypertension [58] and HIV [59] demonstrated more precise reporting of behavioral adherence with less over-reporting in a 30-day period, compared to shorter intervals. This suggested that provider

documentation of medication adherence at monthly maintenance visits with a validated scale (such as visual analogue scale) could be a good screening tool to assess for adherence. Subsequent definitions of nonadherence could be extrapolated from studies of MEMS cap data, providing the 95% adherence cut off.

While there were many studies evaluating adherence rates or prevalence of nonadherence in our population of interest, there were very few studies of interventions to improve adherence. These studies did not have uniform methods of assessing adherence or change in adherence. Growing evidence supports the feasibility and efficacy of technology-based interventions for supporting behavior change in pediatric oncology [60]. In particular, Psihogios et al. [42] demonstrated the feasibility and acceptability of daily text messages as a method of assessing adherence, with convergence on >90% of days with MEMS cap. Together with the pilot study by Bhatia et al. [16] combining daily text reminders with directly supervised therapy, these data suggested a role for daily text reminders as both a tool for measuring and improving adherence to oral maintenance chemotherapy. This was in line with the results of a systematic review of 15 studies of adolescent patients with chronic illnesses reported by Badawy et al. [61] which demonstrated that text messaging and mobile phone app-based interventions had good feasibility and acceptability with moderate evidence for improving adherence. Thakkar et al. [62] further confirmed this in a meta-analysis of 16 randomized clinical trials showing that text messages doubled the odds of medication adherence. However, both authors called for further evaluation of efficacy over longer duration of time. Similarly, with oral maintenance therapy in ALL, further investigation into whether parents and/or adolescents may receive the same benefit from text-based interventions [63] are required, which might lead to stronger habits and higher adherence to medication [64].

*Strengths and Limitations*

Our systematic review had several strengths. First, we followed the recommendations for rigorous systematic review methodology [10,11]. Second, we conducted a comprehensive search—without limits on language or published date, guided by a librarian information specialist—to identify as many relevant studies as possible and included additional hand search by the authors. Finally, two authors independently reviewed each of the included articles.

It is important to note some of the limitations of this systematic review as well. First, although we intended to be comprehensive in our search criteria, it is possible we missed relevant articles. Second, the included studies were very heterogenous in terms of methods of measuring adherence, definitions of nonadherence, analysis of multiple oral chemotherapy agents together versus single specific oral chemotherapy agents, sample size, and age. All this prohibited a meta-analysis from being performed.

## 13. Conclusions

Nonadherence to oral maintenance chemotherapy was prevalent and associated with poor health outcomes. The literature indicated a variety of methods of evaluating adherence with variable rates of adherence. Further rigorous longitudinal studies will be needed to determine optimal strategies for monitoring adherence in clinical practice—and to evaluate potential interventions, thereby promoting medication adherence among children and AYA with ALL and their parents and associated health outcomes.

**Funding:** This project was supported by grant (K23HL150232, PI: Badawy) from the National Heart, Lung, and Blood Institute of the National Institutes of Health. The content is solely the responsibility of the authors and does not necessarily represent the National Institutes of Health. This project was also supported by the NUCATS Voucher Program award for student research.

**Conflicts of Interest:** The authors declare no conflict to interest.

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
