# Peer review of "Adherence to Oral Chemotherapy in Acute Lymphoblastic Leukemia during Maintenance Therapy in Children, Adolescents, and Young Adults: A Systematic Review"

_curroncol, doi:10.3390/curroncol30010056_

Round 1

Reviewer 1 Report

The manuscript is a systemic review described adherence to oral chemotherapy in children, adolescents, and young adults with acute lymphoblastic leukemia during maintenance therapy. This paper is well written and interesting. 

Section "Reasons for nonadherence": Whether hepatotoxicity was nonadherence of maintenance treatment in adult patients only? In my practice, I often observe hepatotoxicity in children (grade 3).

Author Response

Reviewer #1:

The manuscript is a systemic review described adherence to oral chemotherapy in children, adolescents, and young adults with acute lymphoblastic leukemia during maintenance therapy. This paper is well written and interesting. 

  1. Comment: “Section "Reasons for nonadherence": Whether hepatotoxicity was nonadherence of maintenance treatment in adult patients only? In my practice, I often observe hepatotoxicity in children (grade 3).”

Response: Thank you for your comment. We agree that hepatotoxicity is frequently observed in pediatric patients during maintenance therapy for acute lymphoblastic leukemia and may lead to treatment interruptions. 6MP and MTX are known hepatotoxic medications. For our review, we were specifically interested in medication adherence and prevalence of nonadherence. We cite the definition of adherence based on Osterberg et al (Ref 9) in which “medication adherence can be defined as taking medications exactly as prescribed by a medical provider”. In the face of grade 3 and grade 4 hepatotoxicity in our pediatric patients, we apply recommendations provided in the Children’s Oncology Group acute leukemia protocols of dose adjustments and holding of maintenance 6MP and MTX depending on level of toxicity. We did not consider these treatment interruptions as medication nonadherence as they are facilitated and prescribed by a medical provider. In our review, we did not identify any publication that specifically evaluated the role of hepatotoxicity in medication adherence in either pediatric or adult populations. In assessing the reasons for nonadherence, several publications performed qualitative surveys or interviews with subjects to gain insight. Across these publications, hepatotoxicity was not one of the reasons subjects identified as contributing to nonadherence in pediatric patients. Mancini et al (Ref 39), in their survey of adult patients did find that hepatic side effects were identified as response in 2/9 subjects and linked to complete drop out of treatment but did not elaborate further.

Changes in the manuscript: Subsection Reasons for nonadherence (pg 10) “hepatic side effects were associated with nonadherence in adult patients in the form of complete discontinuation of treatment”

Reviewer 2 Report

Title: Adherence to Oral Chemotherapy in Acute Lymphoblastic Leukemia During Maintenance Therapy in Children, Adolescents, and Young Adults: A Systematic Review: Xiaopei L. Zeng ***** , Mallorie B. Heneghan , Sherif M. Badawy

Zeng et al. examine the role of chemotherapeutic adherence in Acute Lymphoblastic Leukemia in this manuscript. Lack of adherence to oral chemotherapy is significant, as is the accompanying risk of relapse. This is an informative analysis of the problem, but it can be enhanced in various ways.

  1. Did the authors include Web of Science database? If not authors need to explain
  2. The authors need to add more details to “Additional articles through hand search were added during the review process.”
  3. The writers must add new results from the previous year ( Refer - An updated literature search was completed in October 2021).
  4. The authors must include the associated keywords with their searches.
  5. The authors must include a methodology for calculating the cumulative incidence of relapse.

Author Response

Reviewer #2:

Zeng et al. examine the role of chemotherapeutic adherence in Acute Lymphoblastic Leukemia in this manuscript. Lack of adherence to oral chemotherapy is significant, as is the accompanying risk of relapse. This is an informative analysis of the problem, but it can be enhanced in various ways.

1. Comment: Did the authors include Web of Science database? If not authors need to explain

Response: Thank you for your question. We did not include Web of science database in our search. The databases included in our search are noted in the article retrieval section (pg 2). We believed that these databases encompassed a wide selection of articles from various disciplines publishing on the topic of medication adherence in leukemia maintenance therapy. We did not additionally include Web of Science because we, under the guidance of our medical librarian, felt that there would be significant overlap with a Web of Science search and the likelihood of missing an impactful article would be exceedingly low.

2. Comment: “The authors need to add more details to ‘Additional articles through hand search were added during the review process.’”

Response: Thank you for your comment. We performed an additional search of PubMed separate from the medical librarian search using key words using keywords “adherence OR compliance” AND “leukemia” AND “maintenance chemotherapy”

Changes in the manuscript: We adjusted the sentence in the Article retrieval subsection (pg 2): “Additional articles through hand search of PubMed were added during the review process using the following keywords: adherence, compliance, leukemia, maintenance chemotherapy.”

3. Comment: “The writers must add new results from the previous year ( Refer - An updated literature search was completed in October 2021).”

Response: Thank you for your suggestion. We performed an updated literature search using the same keywords as our hand search on PubMed in December 2022 prior to the submission of our review, which did not yield any new articles that met our inclusion criteria. Specifically Tang et al and Camire-Bernier et al were both qualitative studies without a specific measure of adherence.

  • Tang, N,Kovacevic, A,  Zupanec, S, et al.  Perceptions of parents of pediatric patients with acute lymphoblastic leukemia on oral chemotherapy administration: A qualitative analysis. Pediatr Blood Cancer. 2022; 69:e29329 https://doi.org/10.1002/pbc.29329.
  • Camiré-Bernier É, Nidelet E, Baghdadli A, Demers G, Boulanger MC, Brisson MC, Michon B, Lauzier S, Laverdière I. Parents' Experiences with Home-Based Oral Chemotherapy Prescribed to a Child Diagnosed with Acute Lymphoblastic Leukemia: A Qualitative Study. Curr Oncol. 2021 Nov 1;28(6):4377-4391. doi: 10.3390/curroncol28060372.

Changes in manuscript: We added the following update to the Article Retrieval section (pg 2) “An updated literature search on PubMed using the keywords from previous hand search was completed in December 2022.”

4. Comment: “The authors must include the associated keywords with their searches.” (pg 2)

Response: Thank you for your suggestion. We previously included keywords included in our medical librarian search (in Article Retrieval subsection pg 2). We have now added the keywords used in our hand search and referenced the same keywords in our updated literature search in December 2022.

Changes in manuscript: The following keywords were included in the sentence referencing our hand search “adherence, compliance, leukemia, maintenance chemotherapy” and in reference the updated literature search in December 2022 “an updated literature search on PubMed using keywords from previous hand search was completed in December 2022.” (pg 2)

5. Comment: The authors must include a methodology for calculating the cumulative incidence of relapse.

Response: Thank you for your comment. The cumulative incidence of relapse we reference in our review was one of the reported outcomes in several of the articles (ref 13-15, 23). Since we did not calculate the CIR ourselves, we did not feel it necessary to include the calculation as part of our methodology. Per Bhatia et al (ref 13), “the cumulative incidence of relapse (at any site) was calculated, treating death as a competing risk” and they cite Gooley et al

  • Gooley TA, Leisenring W, Crowley J, Storer BE. Estimation of failure probabilities in the presence of competing risks: new representations of old estimators. Stat Med. 1999 Mar 30;18(6):695-706. doi: 10.1002/(sici)1097-0258(19990330)18:6<695::aid-sim60>3.0.co;2-o.

Round 2

Reviewer 2 Report

The authors have satisfactorily addressed the comments.